# The ‘Reverse FDUF’ Mechanism of Atrial Excitation–Contraction Coupling Sustains Calcium Alternans—A Hypothesis

**DOI:** 10.3390/biom13010007

**Published:** 2022-12-20

**Authors:** Kathrin Banach, Lothar A. Blatter

**Affiliations:** 1Department of Internal Medicine/Cardiology, Rush University Medical Center, Chicago, IL 60612, USA; 2Department of Physiology & Biophysics, Rush University Medical Center, Chicago, IL 60612, USA

**Keywords:** AP, action potential, APD, AP duration, AR, alternans ratio, [Ca]_i_, cytosolic calcium concentration, CaT, Ca transient, CIRC, Ca-induced Ca release, CRU, Ca release unit, ECC, excitation–contraction coupling, FDUF mechanism, fire-diffuse-uptake-fire mechanism, LCC, voltage-gated L-type Ca channel, RyR, ryanodine receptor, SR, sarcoplasmic reticulum, j-SR, junctional SR, nj-SR, non-junctional SR

## Abstract

Cardiac calcium alternans is defined as beat-to-beat alternations of Ca transient (CaT) amplitude and has been linked to cardiac arrhythmia, including atrial fibrillation. We investigated the mechanism of atrial alternans in isolated rabbit atrial myocytes using high-resolution line scan confocal Ca imaging. Alternans was induced by increasing the pacing frequency until stable alternans was observed (1.6–2.5 Hz at room temperature). In atrial myocytes, action potential-induced Ca release is initiated in the cell periphery and subsequently propagates towards the cell center by Ca-induced Ca release (CICR) in a Ca wave-like fashion, driven by the newly identified ‘fire-diffuse-uptake-fire’ (FDUF) mechanism. The development of CaT alternans was accompanied by characteristic changes of the spatio-temporal organization of the CaT. During the later phase of the CaT, central [Ca]_i_ exceeded peripheral [Ca]_i_ that was indicative of a reversal of the subcellular [Ca]_i_ gradient from centripetal to centrifugal. This gradient reversal resulted in a reversal of CICR propagation, causing a secondary Ca release during the large-amplitude alternans CaT, thereby prolonging the CaT, enhancing Ca-release refractoriness and reducing Ca release on the subsequent beat, thus enhancing the degree of CaT alternans. Here, we propose the ‘reverse FDUF’ mechanism as a novel cellular mechanism of atrial CaT alternans, which explains how the uncoupling of central from peripheral Ca release leads to the reversal of propagating CICR and to alternans.

## 1. Introduction

Alternans is defined as beat-to-beat alternations in action potential duration (APD), contraction strength and Ca transient (CaT) amplitude at a constant heart rate [1,2,3,4,5,6,7,8], and describes a condition that can create temporal and spatial electrophysiological and contractile heterogeneity that facilitates cardiac arrhythmia, including atrial fibrillation [9,10,11]. Atrial CaT alternans shows a unique cellular spatio-temporal organization with complex subcellular dynamics and [Ca]_i_ gradients [7,12,13]. These gradients are directly related to the unique features of Ca release during atrial excitation–contraction coupling (ECC) and are the result of a lack or the irregular organization of the transverse tubule membrane system [14,15], a unique atrial ion channel endowment [16] and atria-specific AP morphology. Atrial myocytes have two types of sarcoplasmic reticulum (SR) Ca stores, peripheral junctional (j-SR) and central non-junctional (nj-SR) SR. Both participate in ECC through active Ca release [15,17,18,19], which results in spatially and temporally inhomogeneous [Ca]_i_ elevation during ECC [6,14,20]. Ca release during ECC is initiated by Ca-induced Ca release (CICR) from the subsarcolemmal array of Ca-release sites of the j-SR triggered by AP-dependent Ca entry through voltage-gated L-type Ca channels (LCCs). The peripheral elevation of [Ca]_i_ gives rise to centripetal CICR propagation from j-SR to the most peripheral array of Ca-release sites of the nj-SR, followed by centripetal CICR propagation through the nj-SR network. Ca release from nj-SR Ca-release sites is facilitated by the ‘fire-diffuse-uptake-fire’ (FDUF) mechanism identified by our group [21,22,23]. The FDUF mechanism entails a tandem activation of the ryanodine receptor (RyR) Ca-release channel by cytosolic and luminal Ca. Through the FDUF mechanism, the SR Ca ATPase (SERCA pump) generates a brief elevation of luminal [Ca]_SR_ at the propagation wave front, which sensitizes the channel to cytosolic CICR and lowers the threshold for its Ca-dependent activation. Atrial myocytes show a higher propensity to develop CaT alterans compared to ventricular myocytes [6,12,13,24,25,26], however, the underlying mechanism for this enhanced pro-arrhythmic vulnerability is poorly understood. Here, we propose, based on new experimental evidence, a hypothesis for a novel CaT alternans mechanism in atrial tissue which synthesizes the unique atrial ECC and Ca-release features with the atria-specific CaT alternans properties. We demonstrate how dysregulation of the previously established FDUF mechanism of atrial ECC is responsible for sustained CaT alternans through a process termed the ‘reverse FDUF’ mechanism.

## 2. Materials and Methods

### 2.1. Myocyte Isolation

Left atrial myocytes were isolated from male New Zealand White rabbits (2.5–2.9 kg; Envigo, Indianapolis, IN, USA). Rabbits were anaesthetized by the intravenous injection of pentobarbital sodium (100 mg/kg). Hearts were excised, mounted on a Langendorff apparatus, and perfused with a nominally Ca-free Tyrode solution for 5 min followed by a minimal essential medium Eagle (MEM) solution containing 20 µM Ca and 22.5 µg/mL Liberase Blendzyme TH (Roche Applied Science, Indianapolis, IN, USA) for 20 min at 37 °C. The left atrium was removed from the heart and digested for an additional 5 min in the enzyme solution at 37 °C. Atrial tissue was minced, filtered, and washed in an MEM solution containing 50 µM Ca and 10 mg/mL of bovine serum albumin. Isolated cells were kept in an MEM solution with 50 µM Ca at room temperature (20–24 °C) until indicator dye loading and subsequent experimentation. Before experimentation, extracellular [Ca] was gradually increased to 2 mM. All protocols were approved by the Institutional Animal Care and Use Committee of Rush University Chicago (22-027; approved 9 May 2022) and complied with the Guide for the Care and Use of Laboratory Animals of the National Institutes of Health.

### 2.2. Solutions and Chemicals

All chemicals were obtained from Sigma-Aldrich (St. Louis, MO, USA) unless stated otherwise. The fluorescent calcium indicator Fluo-4/AM was obtained from Thermo Fisher Scientific (Waltham, MA, USA). The Tyrode solution contained (in mM): 130 NaCl, 4 KCl, 2 CaCl_2_, 1 MgCl_2_, 10 D-glucose, 10 HEPES; pH 7.4 with NaOH. All cells were plated on laminin-coated (1–2 mg/mL) glass coverslips and superfused with 2 mM Ca Tyrode solution.

### 2.3. Confocal [Ca]_i_ Measurements

For confocal [Ca]_i_ measurements, cells were loaded with Fluo-4/AM (10 µM) for 20 min, followed by a washing period (>10 min) in the Tyrode solution. Fluo-4 was excited at 488 nm and fluorescence emission was recorded at >515 nm. Confocal line scan images were recorded at 512 lines/s with a Nikon A1R system (Nikon Corporation, Melville, NY, USA) using a 60× oil-immersion objective lens (NA = 1.49). The scan line (pixel dimension 0.03 µm) was placed along the transverse axis of the cell avoiding the nucleus (Figure 1). Subcellular CaTs reflecting Ca release from j-SR (SS, subsarcolemmal) and nj-SR (CT, central) were derived from 1–1.5 µm wide regions. Action potentials and CaTs were elicited by the electrical field stimulation of intact atrial myocytes using a pair of platinum electrodes (voltage set at ~50% above the threshold for contraction). Changes in [Ca]_i_ are expressed as F/F_0_ where F represents background-subtracted time-dependent Fluo-4 fluorescence and F_0_ refers to diastolic Fluo-4 fluorescence levels measured under control steady-state conditions during electrical stimulation at the beginning of a recording. Experiments were conducted at room temperature (20–24 °C).

### 2.4. CaT Alternans

CaT alternans was induced by incrementally increasing the pacing frequency until stable CaT alternans was observed (1.6–2.5 Hz was the typical frequency range for stable CaT alternans). The degree of CaT alternans was quantified as the alternans ratio (AR). AR = 1 − [Ca]_i,Small_/[Ca]_i,Large_, where [Ca]_i,Large_ and [Ca]_i,Small_ are the amplitudes of the large and small CaTs of a pair of alternating CaTs. By this definition, AR values fall between 0 and 1, where AR = 0 indicates no CaT alternans and AR = 1 indicates a situation where SR Ca release is completely abolished on every other beat. CaTs were considered to be alternating when the beat-to-beat difference in CaT amplitude exceeded 10% (AR > 0.1) [26].

## 3. Results and Discussion

### 3.1. A Unique Atrial ECC Feature: The FDUF Mechanism

Excitation–contraction coupling is strikingly different in atrial compared to ventricular myocytes. Their structural differences and unique endowment with ion channels [16] have profound consequences for AP morphology and Ca release. In atrial cells, the transverse tubular membrane system is poorly developed or lacking [14,15,27,28], with rabbit atrial myocytes representing an example of a complete lack of t-tubules [21,23,29]. The complete lack of t-tubules indicates that in rabbit atrial myocytes, j-SR and nj-SR Ca release are spatially clearly separated. Therefore, the SR—the intracellular Ca store—only makes junctions with the surface membrane in the cell periphery. This junctional SR forms Ca-release units (CRUs) [30] that are organized as peripheral couplings [19,31] and are reminiscent of the ‘classical’ ventricular couplon [32] (characterized by the close spatial association of LCCs and RyR Ca-release channels). Different from ventricle, however, atrial myocytes have abundant nj-SR distant from the surface membrane, and both j-SR and nj-SR CRUs participate in atrial ECC through active RyR Ca release [15,17,18,19,33]. Due to the lack of t-tubules in rabbit atrial myocytes, the Ca releases from j-SR and nj-SR are spatially clearly separated. In fact, the main burden to deliver the Ca necessary for atrial contraction is shouldered by the nj-SR. In stark contrast to ventricular cells, AP-induced Ca release in atrial cells is spatially inhomogeneous [6,14,20] (Figure 1A). The rate of [Ca]_i_ rise is fastest in the cell periphery, and the amplitude of the junctional CaT (CaT_j_) typically exceeds the non-junctional CaT (CaT_nj_). Atrial Ca release occurs by the following ECC mechanism. The AP activates Ca entry through LCCs, however, the entering of Ca triggers CICR only from j-SR RyRs. The elevated peripheral [Ca]_i_ subsequently propagates in a Ca wave-like fashion centripetally (between nj-SR CRUs) into the cell’s interior. The nj-SR CRUs never ‘see’ a fast, local LCC-driven Ca trigger, instead, CICR is triggered by a slower and diffuse increase in [Ca]_i_. Yet, a normal atrial beat-to-beat operation critically relies on robust nj-SR CRU activation and an efficient CICR propagation. This has long been a quandary, considering that RyR Ca-sensitivity is normally low [34] and the bulk cytosolic CaT amplitude barely exceeds 1 µM. We recently made a paradigm-shifting discovery towards unraveling the hitherto unknown mechanism underlying this paradoxical nj-SR CICR efficiency by proposing the ‘fire-diffuse-uptake-fire’ (FDUF) mechanism of atrial ECC [21,22,23]. Inspired by our previous work on Ca wave mechanism [35], we demonstrated that nj-SR Ca release is facilitated by a transient intra-SR Ca sensitization signal. Specifically, at the propagating Ca wave front, the SR SERCA pump takes up Ca into the SR and establishes a transient local [Ca]_SR_ elevation, which, via luminal RyR sensitization, lowers the threshold for cytosolic CIRC. We identified this ‘tandem’ mode of RyR luminal/cytosolic regulation as a novel atrial ECC paradigm, the FDUF mechanism, and determined thereby a novel role of SERCA, in addition to its well-established function of refilling the SR.

### 3.2. Hypothesis: The ‘reverse FDUF’ Mechanism of Atrial Alternans

Alternans is a recognized risk factor for cardiac arrhythmia—including atrial fibrillation (AF) [9,10,11]—and sudden cardiac death [36,37,38,39]. At the cellular level, cardiac alternans is defined as beat-to-beat alternations in contraction amplitude (mechanical alternans), AP duration (APD or electrical alternans), and CaT amplitude (CaT alternans) at a constant stimulation frequency. Its cause is multifactorial (for reviews see [1,2,3,4,5,6,7,8]). CaT and APD alternans are typically highly synchronized and the regulation of [Ca]_i_ and membrane potential (V_m_) is bi-directionally coupled and governed by complex overlapping feedback pathways. Atrial alternans can be V_m_- or Ca-driven; however, recent progress (including our own [26,40]) and computational findings increasingly point to Ca signaling disturbances as the primary cause of alternans [41,42,43,44]. Atrial myocytes are particularly susceptible to CaT alternans [6,12,13,24,25,26], which, as we hypothesize, is a result of the unique features of atrial ECC, especially the subcellular [Ca]_i_ gradients and the FDUF mechanism. At the cellular level, atrial CaT alternans is distinctly different from ventricular alternans. We have shown previously that atrial Ca alternans is subcellularly inhomogeneous [7,12,13] with transverse and longitudinal gradients of the degree of CaT alternans, and even subcellular regions that alternate out of phase. The latter generates a substrate for spontaneous (i.e., not electrically triggered) proarrhythmic Ca release, implying a mechanistic link to atrial arrhythmia at the cellular level [13]. The key mechanism that sustains robust atrial Ca release is the atrial FDUF mechanism. It guarantees a tight functional coupling between j-SR Ca release and the subsequent directional (centripetal) propagation of nj-SR Ca release, where the CaT_j_ initiates the CaT_nj_. Based on novel experimental observations (Figure 1B and Figure 2), we propose here the hypothesis that the deregulation of the FDUF mechanism and the uncoupling of nj-SR from j-SR Ca release is a major mechanism that is responsible for the development of atrial alternans. By uncoupling, we refer here to a violation of the strict sequence of CaT_j_ triggering CaT_nj_. During normal ECC (Figure 1A), the amplitude of CaT_j_ exceeds CaT_nj_ (ratio CaT_j_/CaT_nj_ > 1; Figure 1A) and ensures the robust propagation of Ca release from the periphery to the cell center, thereby sustaining nj-SR Ca release with high fidelity. However, new data show that during ECC, the subcellular [Ca]_i_ gradient can reverse (CaT_j_/CaT_nj_ < 1). Under these circumstances, Ca release, facilitated by the reverse FDUF mechanism, propagates back to the cell periphery and enhances subsarcolemmal [Ca]_i_. This back propagation of Ca release represents a violation of the normal sequence of events of ECC and is referred to here as the uncoupling of j-SR and nj-SR Ca release. This reversal of the subcellular [Ca]_i_ gradient (CaT_j_/CaT_nj_ < 1) during the late phase of the CaT results in the onset of alternans, is sustained during the course of CaT alternans, and increases the alternans ratio (Figure 1B). Furthermore, the reversal of the subcellular [Ca]_i_ gradient correlated with the occurrence of a secondary Ca-release component during the late phase of the alternans CaT (marked by arrow heads in Figure 2). We analyzed a set of eight atrial myocytes from three different rabbits where CaT alternans was induced by electrical pacing. In seven of the eight cells analyzed in detail, a reversal of the subcellular [Ca]_i_ gradient during the development of alternans was identified, and all eight cells revealed secondary Ca-release components during alternans.

The reverse FDUF mechanism contributes to CaT alternans through three potential mechanisms. First, the secondary Ca-release component (Figure 2) that occurs with latency during the late phase of the alternans CaT further enhances the amount of Ca released during the large-amplitude alternans CaT. This secondary SR Ca-release component enhances the degree of CaT alternans and increases the alternans ratio. Second, the boosting of Ca release through the reverse FDUF-dependent secondary Ca-release component empties the SR to a larger degree, and incomplete refilling during the subsequent diastole results in a smaller CaT amplitude during the next beat, also resulting in an increased alternans ratio. End-diastolic alternations in SR Ca content has indeed been proposed as an alternans mechanism [4,43,45,46]; however, this notion is a matter of debate and it is noteworthy that CaT alternans can be sustained without alternations in the beat-to-beat filling of SR [12,40]. Third, the larger and prolonged release of Ca during the large-amplitude alternans CaT brought upon by the reverse FDUF mechanism leads to the prolonged refractoriness of SR Ca release, thereby attenuating the Ca release and CaT amplitude during the next beat, again further increasing the CaT alternans ratio. We [40] and others [47,48] have shown previously that the beat-to-beat alternation in SR Ca-release refractoriness is indeed a major contributor to the alternans mechanism. In summary, we have identified three potential mechanisms through which the reverse FDUF mechanism hypothesis facilitates CaT alternans in atrial myocytes.

## 4. Conclusions

We forward the hypothesis that the reverse FDUF mechanism is responsible for sustained CaT alternans in the atrium. The key elements of the reverse FDUF mechanism are the uncoupling of nj-SR Ca release from j-SR release and the reversal of subcellular [Ca]_i_ gradients during ECC. As a consequence, this enhances the large-amplitude alternans CaT through additional Ca release and prolongs the refractoriness of SR Ca release. The latter curtails Ca release during the subsequent small-amplitude alternans CaT, and thus sustains CaT alternans.

The findings presented here also have therapeutic ramifications for atrial arrhythmia. The FDUF mechanism of atrial ECC and Ca alternans is critically dependent on the action of the SERCA pump which generates the intra-SR Ca sensitization signal required for the tandem activation of the RyR Ca-release cannel by luminal and cytosolic Ca. SERCA indeed has been implicated in alternans [49,50,51] and in the alternans–AF link [52,53]. Thus, targeting SERCA opens the possibility of novel therapeutic strategies of atrial alternans prevention, and thus, the suppression of a pro-arrhythmic condition directly related to atrial fibrillation, the most common form of cardiac arrhythmia. By identifying the FDUF mechanism as critical for atrial Ca alternans, we have further characterized SERCA as an anti-arrhythmic target.

## Figures and Tables

**Figure 1 biomolecules-13-00007-f001:**
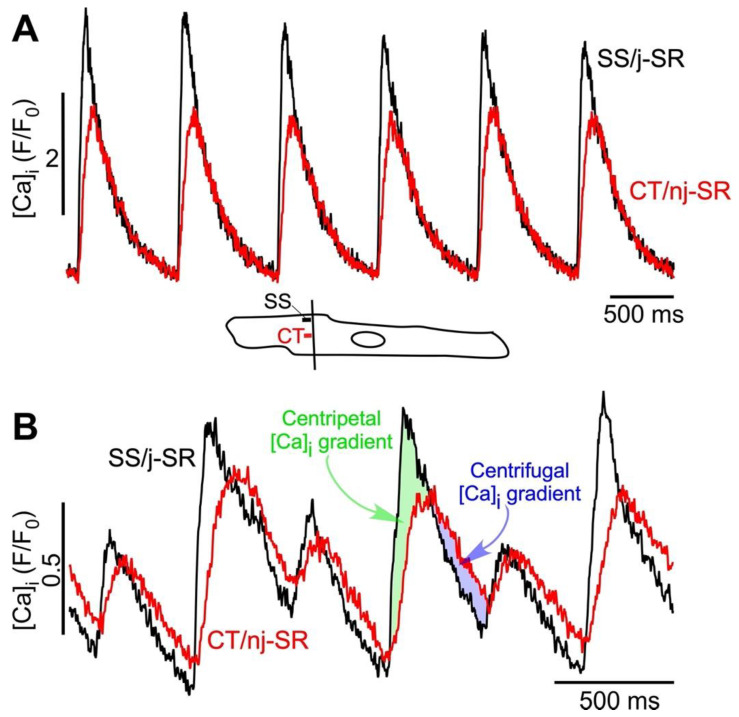
Atrial ECC and ‘reverse FDUF’ mechanism. (**A**) Subsarcolemmal (SS) Ca release from junctional SR (j-SR) and from central (CT) non-junctional SR (nj-SR) during electrical pacing at 1.3 Hz in the absence of CaT alternans. The Ca release from j-SR is faster and reaches a higher amplitude than nj-SR Ca release. CT [Ca]_i_ never exceeds SS [Ca]_i_. Bottom: confocal scan line position and SS and CT regions of interest. Scan direction perpendicular to the longitudinal axis of the cell (transverse scan). (**B**) SS and CT Ca release during stable alternans. Alternans was induced by electrical pacing at 2.5 Hz. The Ca release from j-SR is faster and reaches a higher amplitude than nj-SR Ca release; however, during the later phase of CaT, the CT CaT_nj_ amplitude exceeds SS CaT_j_ and reverses the subcellular [Ca]_i_ gradient from centripetal to centrifugal.

**Figure 2 biomolecules-13-00007-f002:**
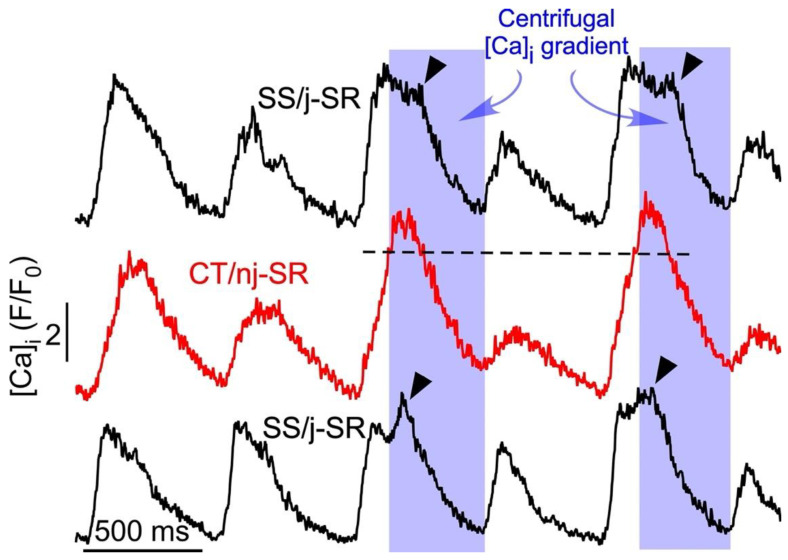
Secondary SR Ca release during atrial CaT alternans. Subcellular CaT traces recorded from 1 µm wide regions by confocal transverse line scan imaging from the cell center (CT/nj-SR; red trace) and opposite subsarcolemmal regions (SS/j-SR; black traces). Alternans was induced by electrical pacing at 1.8 Hz. Reversal of the subcellular [Ca]_i_ gradient from centripetal to centrifugal (shaded) during the later phase of the CaT elicits secondary SS j-SR Ca release (arrow head), leading to a prolongation of the SS CaT and enhanced refractoriness of j-SR Ca release during the subsequent beat. Dashed line indicates peak SS CaT.

## Data Availability

The data presented in this study are available on request from the corresponding author.

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
