# Peer review of "The ‘Reverse FDUF’ Mechanism of Atrial Excitation–Contraction Coupling Sustains Calcium Alternans—A Hypothesis"

_biomolecules, 2022, doi:10.3390/biom13010007_

Round 1
Reviewer 1 Report
In this study, Banach & Blatter put forward a new hypothesis on a calcium signaling mechanism in atrial myocytes that sustains calcium alternans. The mechanism is termed "reverse fire-diffuse-uptake-fire" and builds on a recent publication by the same group, in which they described the "fire-diffuse-uptake-fire" mechanism for calcium propagation during normal excitation-contraction coupling in atrial myocytes. Calcium alternans is a proarrhythmic mechanism that has been implicated in cardiac arrhythmias. The topic is highly relevant and of interest for both basic and clinical researchers.
In general, the paper is of high quality, experiments and figures are sound and illustrate the main findings in a clear fashion. I have only a few comments:
1) In figure 1B, you show that there is a reversal of the calcium gradient during calcium alternans. It would be nice to know the number of cells (or the fraction of cells) in which such a pattern was observed. Maybe you could add this quantitative information as new panel figure 1.
2) The observation of a reversal of the calcium gradient (from centripetal to centrifugal) during alternans per se is not compelling evidence for a reversal of active calcium release. For example, during normal excitation-contraction coupling some atrial myocytes show a similar reversal of the calcium gradient (see figure 1D in Huser et al, 1996. J. Physiol. 494;641) and in this case this has been interpreted simply as active calcium propagation from the periphery to the center of the cell (without a reversal of active calcium release). Maybe you can elaborate in a bit more detail on this issue.
3) Figure 2 is a nice example for the proposed reverse fire-diffuse-uptake-fire mechanism. Was this a common finding during alternans? Again, some more quantitative information would be appreciated.
4) In the publication of Maxwell & Blatter (2017. J. Physiol. 595) on the fire-diffuse-uptake-fire mechanism, nice recordings of SR calcium changes during excitation-contraction coupling have been presented. Do you happen to have similar recordings of SR calcium in the case of calcium alternans, which would support the findings shown in figure 2 here?
Reviewer 2 Report
The paper discusses important issue. The paper in principle is well written, and well designed, but not so well presented. The abstract is absolutely unacceptable, since does not contain any sentences about the original experiments. Please rewrite.
Reviewer 3 Report
In their paper Banach and Blatter address the mechanism of Ca alternans in atrial myocytes. Using spatially resolved recording of pacing-induced Ca transients in rabbit atrial myocytes authors conclude that the major mechanisms of Ca alternans in atrial myocytes is a deregulation of the fire-diffuse-release-fire (FDUF) mechanism and the uncoupling of non-junctional (njCR) from junctional (jCR) SR Ca release.
Although it is possible that uncoupling of jCR from njCR contributes to the generation of Ca alternans, it is impossible to confirm that this is a major mechanism of Ca alternans in atrial myocytes based on the data presented in the manuscript. Traces presented in fig.2 as jCR display combination of fast and slow onset of Ca transient suggesting that presence of njCR. More rigorous separation of jCR from njCR is suggested (for example based on latency between electrical stimulation and maximal rate of Ca release PMID: 26103619).
Quantitative analysis of coupling efficiency between jCR and njCR (e.g. PMID: 34512394) and statistical analysis of its relationship with the Ca alternans will help to prove the main point on the major mechanism.
Discussion of the potential mechanisms of uncoupling and deregulation of FDUF will be helpful.
The role of ‘reverse FDUF’ mechanism in sustained Ca alternans requires some experimental evidence. It seems that ‘reverse FDUF’ is a regenerative Ca—induced Ca release propagated from the regions where junctional and non-junctional sites were coupled to the regions where they were uncoupled (as illustrated in fig.3 in PMID: 33398498). These Ca waves are known to underlie subcellular Ca alternans. However, their role in pathogenesis of global Ca alternans has to be established. I believe that analysis of subcellular patterns of Ca release in XY-scans before the onset and during sustained Ca alternans will provide more convincing argument for the author’s point..
Are the mechanisms proposed here for the Ca alternans unique for atrial myocytes or they can also be applied to ventricular myocytes of large animals including humans as well as diseased ventricular myocytes that are known to have less dense t-tubule system, similar to that of the atrial myocytes?
Round 2
Reviewer 2 Report
The authors answered all my oroginal concerns. I suggest now acceptance.
Reviewer 3 Report
no further comments